# LANGUAGE MODELS TRAINED TO DO ARITHMETIC PREDICT HUMAN RISKY AND INTERTEMPORAL CHOICE

**Jian-Qiao Zhu**
Department of Computer Science
Princeton University
`jz5204@princeton.edu`

**Haijiang Yan**
Department of Psychology
University of Warwick

**Thomas L. Griffiths**
Department of Psychology and Computer Science
Princeton University

## ABSTRACT

The observed similarities in the behavior of humans and Large Language Models (LLMs) have prompted researchers to consider the potential of using LLMs as models of human cognition. However, several significant challenges must be addressed before LLMs can be legitimately regarded as cognitive models. For instance, LLMs are trained on far more data than humans typically encounter, and may have been directly trained on human data in specific cognitive tasks or aligned with human preferences. Consequently, the origins of these behavioral similarities are not well understood. In this paper, we propose a novel way to enhance the utility of language models as cognitive models. This approach involves (i) leveraging computationally equivalent tasks that both a language model and a rational agent need to master for solving a cognitive problem and (ii) examining the specific task distributions required for a language model to exhibit human-like behaviors. We apply this approach to decision-making – specifically risky and intertemporal choice – where the key computationally equivalent task is the arithmetic of expected value calculations. We show that a small language model pretrained on an ecologically valid arithmetic dataset, which we call Arithmetic-GPT, predicts human behavior better than many traditional cognitive models. Pretraining language models on ecologically valid arithmetic datasets is sufficient to produce a strong correspondence between these models and human decision-making. Our results also suggest that language models used as cognitive models should be carefully investigated via ablation studies of the pretraining data.

## 1 INTRODUCTION

Scientists studying the behavior of Large Language Models (LLMs) in cognitive tasks typically performed by humans have found substantial evidence that LLMs produce performance similar to that of human participants (Binz & Schulz, 2023b; Horton, 2023; Zhu & Griffiths, 2024; Dasgupta et al., 2022; Frank, 2023b; Marjieh et al., 2023; Webb et al., 2023; Coda-Forno et al., 2024).[1] Like humans, LLMs often make judgments and decisions that deviate from rational norms (Binz & Schulz, 2023b; Horton, 2023; Zhu & Griffiths, 2024; Coda-Forno et al., 2024). For instance, GPT-3 demonstrates human-like biases in risky choice, such as risk aversion and loss aversion (Binz & Schulz, 2023b), and the statistical properties of probability judgments generated by LLMs align qualitatively with those of humans (Zhu & Griffiths, 2024). LLMs also make errors in other related settings; for example, when trained on the task of predicting the next token in sequences involving arithmetic

---

[1]LLMs can also display significant deviations from human behavior (e.g., Chen et al., 2023; Hagendorff et al., 2023).

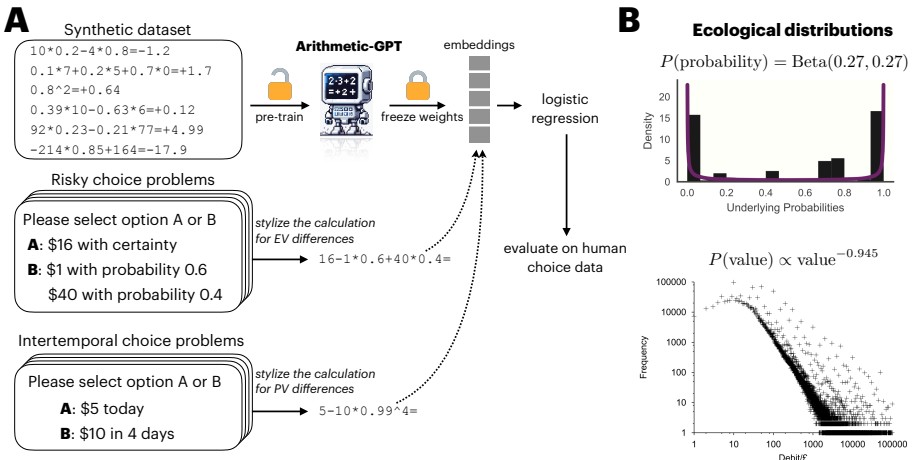

Figure 1: **(A) Pre-training and evaluation pipelines.** We began by generating a synthetic dataset comprised of mathematical equations including addition, subtraction, multiplication, and exponentiation. Arithmetic-GPT was pretrained on this synthetic dataset. After training, we froze model weights and extracted embeddings from the pretrained model, which then processes stylized choice tasks as input. These embeddings were subsequently compared with human choice data to evaluate their correspondence. **(B) Ecological distributions of probabilities and values.** In the top panel, English probability-describing phrases (black bars) can be modeled using a $\text{Beta}(0.27, 0.27)$ distribution. In the bottom panel, the value distribution of debits from UK bank accounts (scatterpoints) follows a power-law distribution. Figures adapted from Zhu et al. (2020) and Stewart et al. (2006).

operations, they fail to learn precise arithmetic operands and instead approximate the correct results up to a certain input range (Nogueira et al., 2021; Lee et al., 2023).

In this paper, we focus on risky and intertemporal choice as exemplar domains for comparing the behavior of language models and humans (see Figure 1). Central to both domains is the computational challenge of calculating expectations. To assess the benefits of engaging in a gamble, an intelligent system must be able to calculate the expected value (EV) of the gamble, typically represented as:

$$EV(A) = \sum_{i \in A} p_i \times x_i \tag{1}$$

where each outcome $i$ of gamble $A$ is associated with a payoff $x_i$ and a probability $p_i$, with the constraint that $\sum_i p_i = 1$. Similarly, in considering an intertemporal choice the computation of the present value (PV) of future outcomes in $A$ is crucial:

$$PV(A) = \sum_{t \in A} d^t \times x_t \tag{2}$$

where the value $x_t$ is realized at time $t$ and is discounted by a factor of $d$, reflecting the time preference of the decision-maker. Note that a risk-neutral and time-consistent agent should always select the option that maximizes EV and PV. However, extensive research in economics and psychology demonstrates that people systematically deviate from this maximizer model (Kahneman & Tversky, 2013; Gigerenzer & Gaissmaier, 2011; Laibson, 1997; Zhu et al., 2020).

Pretrained, off-the-shelf LLMs, such as the GPT and LLaMA series, have demonstrated behavioral similarities to humans in tasks involving risky and intertemporal choices (Horton, 2023; Binz & Schulz, 2023b; Manning et al., 2024). However, the embeddings generated by these LLMs are not by default able to account for human data. For example, embeddings from the LLaMA-1-65B model, when not finetuned on human risky choices, poorly predict those choices (Binz & Schulz, 2023a). Therefore, understanding what enables LLMs to exhibit human-like decision-making behavior remains an unresolved challenge.

We propose a hypothesis about how human-like decision patterns might be produced in language models for risky and intertemporal choice: such human-like behaviors might arise from a lan-

guage model trained on ecologically valid calculations of expectations. This suggests that human-like biases could result from the training task and numerical reasoning, independent of natural language supervision. As a corollary, this hypothesis also implies that deviations from rational choice in humans could be primarily explained by computational errors during the EV or PV calculations. To test this hypothesis, we generate a series of synthetic datasets that contain expected value computations; examples include `0.5*100=+50`, `0.8*1+0.8^2*10=+7.2`, and `30*0.79-261*0.83=-192.93`. Subsequently, we train a small, randomly-initialized language model (approximately 10M parameters) on these datasets. After training, we extract embeddings from the now pretrained language model and analyze how well they can account for human choices.

We conduct carefully ablated experiments on different aspects of the synthetic data to isolate the factors that result in embeddings that better predict human choice patterns. Our findings reveal that when the synthetic dataset reflects the ecological distributions of probabilities and values–mirroring real-world frequencies–the resulting embeddings best predict human choices. With this pretraining, models based on the derived embeddings outperform many existing behavioral models of risky and intertemporal choice.

## 2 BACKGROUND

The question of whether language models can serve as models of human cognition has sparked intense debate within the fields of machine learning and cognitive science (Frank, 2023b; Messeri & Crockett, 2024; Griffiths et al., 2023; Horton, 2023). Although there are behavioral similarities between off-the-shelf LLMs and humans, these do not inherently qualify LLMs as effective cognitive models (c.f. the Clever Hans effect) (Shiffrin & Mitchell, 2023). There are compelling reasons why current LLMs may not be suitable as cognitive models. First, LLMs are trained on datasets vastly larger than those available to human learners (Frank, 2023a). Second, LLMs may have already been trained on the test questions, particularly if the training data is undisclosed and poorly controlled (Jiang et al., 2024). Third, the inclusion of value alignment steps, such as Reinforcement Learning from Human Feedback (Ziegler et al., 2019) and Direct Preference Optimization (Rafailov et al., 2024), may artificially enhance human-like behaviors in leading LLMs. Finally, the immense size of deep neural networks and the proprietary nature of leading LLMs hinder detailed investigation into their internal representations.

Researchers have taken steps to address some of the concerns associated with using LLMs as cognitive models. One branch of research looks at the potential benefits of fine-tuning off-the-shelf LLMs to better understand human cognition. For instance, fine-tuning the LLaMA-1-65B model (Touvron et al., 2023) on datasets of human choices results in a model that can predict human data more accurately than traditional cognitive models (Binz & Schulz, 2023a). Although the specific mechanisms underlying the finetuned LLM's ability to replicate human choices remain unclear, this work serves as a proof-of-concept and suggests that the embeddings learned from extensive pretraining on Internet text and/or from the value alignment process may offer valuable insights into human cognitive processes that complement those provided by traditional cognitive models.

Another line of research emphasizes the importance of the composition of synthetic datasets, which are critical for enhancing certain capabilities of LLMs (Lee et al., 2023). These studies typically assess LLMs based on their problem-solving abilities rather than their human-likeness. For example, it has been found that larger language models tend to perform arithmetic tasks, such as addition and multiplication, better than their smaller counterparts (Yuan et al., 2023). Moreover, the order of input and the inclusion of intermediate information about task decomposition have been shown to facilitate chain-of-thought reasoning, which significantly helps smaller language models in mastering arithmetic tasks (Lee et al., 2023).

Finally, there is a precedent for the idea that pretraining machine learning models on synthetic datasets can improve performance in predicting human decisions. Bourgin et al. (2019) showed that a model pretrained on choice data generated from a psychological theory could perform extremely well in predicting human decisions when fine-tuned with a small amount of human data. Our approach builds on this idea, but reduces it to the most primitive components – rather than pretraining on data generated from a psychological theory, we pretrain on a task that captures the basic computations required to make rational decisions.

## 3 ARITHMETIC-GPT: A SMALL LANGUAGE MODEL TRAINED TO PERFORM ARITHMETIC

In this paper, we confront the challenges of making language models as cognitive models head-on. First, we define a data generation algorithm to produce synthetic datasets, thereby gaining complete control over the training data for language models and addressing issues related to data gaps and contamination. Second, we have direct access to the neural activation patterns that are crucial for decision-making processes. This approach allows us to more thoroughly evaluate and understand the capabilities and limitations of language models.

### 3.1 MODEL DETAILS

Our small language model employs a standard architecture for a Generative Pretrained Transformer (GPT) model (Vaswani et al., 2017; Radford et al., 2019). Detailed specifications of the model architecture are provided in Table 1.

Table 1: Arithmetic-GPT 10M: a small language model pretrained to do arithmetic

| Pre-training hyperparameters | Value |
| --- | --- |
| Hidden size | 320 |
| Layers | 8 |
| Heads | 8 |
| Context length | 26 |
| Vocabulary size | 320 |
| Attention variant | Causal self attention |
| Dropout | 0.2 |
| Biases | None |

**Tokenizer.** To handle a domain-specific vocabulary dedicated for arithmetic equations, we built a custom tokenizer on the sub-word level. The vocabulary size is 320, containing special tokens (e.g., `<AMB>`, `<PAD>`), arithmetic operators (e.g., $+, -, *, ., \hat{}, =$), and all the integers from 0 to 300 which are designed to split numbers into individual digits. The vocabulary can cover most EV calculations in risky choice and PV calculations in intertemporal choice tasks.

**Positional embedding.** Absolute positional embedding was learned during training through an embedding layer. Each position had a corresponding 320 dimensional embedding vector.

**Attention Mask.** To ensure that attention is only applied to the left in the input sequence, we incorporated a causal mask to prevent attending to future tokens when predicting the next one.

### 3.2 SYNTHETIC DATASETS

At the heart of the EV and PV computations lies the multiplication of two real numbers. Typically, each outcome appears in the computation as either a probability multiplied by a value, or a discount factor multiplied by a value. Probabilities are real numbers ranging from 0 to 1, represented with a precision of up to two decimal places. Similarly, values are real numbers that range from 0 to 300, also with a maximum of two decimal places. We selected these ranges to align with the numerical scope of human experimental studies. A single training example involves either the addition or subtraction of two simulated outcomes, which together constitute the left-hand side of an equation. We then compute the corresponding result and place it on the right-hand side of the equation. In total, we randomly simulated 1M such equations.

In our experiment, we evaluate four variants of the data generation algorithm by manipulating the frequency of probabilities and values (see Table 3). The **Uniform synthetic data** generates probabilities and values with maximum uncertainty; probabilities are uniformly distributed between 0 and 1 (i.e., $U[0, 1]$), and values range from 0 to 300 (i.e., $U[0, 300]$). Conversely, the **Ecological synthetic data** generates probabilities following a Beta distribution $\text{Beta}(0.27, 0.27)$ (Zhu et al., 2020) and values according to a power-law distribution with an exponent of $-0.945$ for the same range (Stewart et al., 2006). These distributions are chosen because they have been shown to closely match

the natural frequencies of probabilities and values in real-world scenarios (Stewart et al., 2006; Zhu et al., 2020) (see Figure 1B for details). For both uniform and ecological synthetic datasets, we also created a matching dataset where the answers on the right-hand side are generated with a 50% chance of displaying the incorrect sign (i.e., the ablated variants in Table 3). In each of the four synthetic datasets, we randomly masked 10% of the probability values using a special `<AMB>` token to denote unknown probabilities.

### 3.3 PRETRAINING DETAILS

We trained our Arithmetic-GPT models from scratch with a context length of 26. Batch size was set to 2048 lines of mathematical equations, randomly sampled from the synthetic datasets, and the learning rate was $10^{-3}$. To optionally terminate the training process, we designated 90% of the synthetic dataset as the training set, reserving the remaining 10% for validation. We used cross-entropy loss to measure the discrepancy between the predicted and target sequences. Training was stopped when the validation loss plateaued, with validation loss evaluated every 100 epochs (see Figure A1 in Appendix C for an example learning curve). The AdamW optimizer was used.

### 3.4 HUMAN TARGETS

To investigate whether Arithmetic-GPT contains information pertinent to explaining human decision-making, we reanalyzed recent experiments in which people were asked to make risky and intertemporal choices (Peterson et al., 2021; Erev et al., 2017; Gershman & Bhui, 2020; Agrawal et al., 2023). The primary reason for examining these particular types of human choices is that calculations of expected value are integral to making rational decisions (see Equations 1 and 2). In other words, they are computationally equivalent tasks under the assumption of rationality.

As summarized in Table 2, we sampled four existing datasets from the literature, which included two large-scale experiments (Peterson et al., 2021; Agrawal et al., 2023). In experiments involving risky choices (Peterson et al., 2021; Erev et al., 2017), participants were often presented with two options, each fully describing the details of a gamble (see Figure 1A risky choices). In cases involving ambiguous gambles where probabilities are unknown, we used the special token `<AMB>` to denote gambles with unknown probabilities. We excluded decision-with-feedback trials, as these would require additional assumptions about how individuals respond to feedback (but see Table A2 of Appendix B for a comparison based on the entire `choices13k` dataset). In intertemporal choice tasks (Gershman & Bhui, 2020; Chávez et al., 2017; Agrawal et al., 2023), participants were also presented with two options, typically offering a choice between a smaller, sooner payoff and a larger, later payoff (see Figure 1A intertemporal choices). Without loss of generality, we fixed the annual discount factor at $d_{\text{year}} = 0.85$ throughout the paper. This also corresponds to a monthly discount factor of $d_{\text{month}} = 0.98$ and a daily discount factor of $d_{\text{day}} = 0.99$. We rescaled the values in both options by the same factor to fit within the specified range.

Table 2: Overview of human experiments and data sources

| Paper | Dataset | Domain | No. Participants | No. Problems |
|---|---|---|---|---|
| Peterson et al. (2021) | `choices13k` | Risky choices | 15,153 | 13,006 |
| Erev et al. (2017) | `cpc18` | Risky choices | 446 | 270 |
| Gershman & Bhui (2020) | `gershman20` | Intertemporal choices | 221 | 4,794 |
| Agrawal et al. (2023) | `agrawal23` | Intertemporal choices | 12,906 | 9,853 |

*Note.* In our analysis of the `choices13k` and `cpc18` datasets, we excluded trials involving risky choices made with feedback. Modeling how individuals respond to feedback requires additional cognitive mechanisms beyond the scope of this work.

### 4 OTHER MODELS

We also conduct a model comparison on human data, evaluating the following approaches: (i) classical behavioral models such as prospect theory (Kahneman & Tversky, 2013) and the hyperbolic discounting model (Laibson, 1997); (ii) a neural network directly trained on the human datasets; (iii)

an untrained Arithmetic-GPT model; and (iv) an off-the-shelf, open-weight LLM, LLaMA-3-70B-Instruct.[2]

**Classical behavioral models.** To explain human risky choices, prospect theory proposes an S-shaped utility function that is concave for gains, convex for losses, and steeper for losses than for gains (reflecting loss aversion). The utility function is represented mathematically as:

$$U(x) = \begin{cases} x^\alpha, & \text{if } x \geq 0 \\ -\lambda x^\beta, & \text{if } x < 0 \end{cases} \tag{3}$$

where $x$ denotes the value, while the shape parameters $\alpha$ and $\beta$ define the curvature of the utility function. The parameter $\lambda \geq 1$ reflects loss aversion. The theory further suggests that individuals possess a distorted perception of probabilities, modeled as follows:

$$w(p) = \frac{p^\gamma}{(p^\gamma + (1-p)^\gamma)^{1/\gamma}} \tag{4}$$

where $p$ is the objective probability, and $\gamma$ is a parameter controlling the curvature of the weighting function. Consequently, the utility of a gamble is formally expressed as:

$$U(A) = \sum_{i \in A} w(p_i) \times U(x_i) \tag{5}$$

Contrary to the consistent risk preferences implied by Equation 1 or other monotonic transformations of value, prospect theory suggests that risk preferences are inconsistent across different values and probabilities. This inconsistency results in incoherent choices when individuals are faced with risky decisions (Tversky & Kahneman, 1992).

To capture human time preferences, particularly the impact of present bias, the hyperbolic discounting model suggests that future values should be discounted as follows:

$$PV(x_t) = \frac{x_t}{1 + kt} \tag{6}$$

where $x_t$ is the value to be received at future time $t$, and $k$ is the discount factor that quantifies the degree of time preference. In contrast to the consistent time preferences implied by Equation 2, the hyperbolic discounting model suggests that time preferences are inconsistent across different time horizons, leading to stronger preference to immediate over future rewards (Laibson, 1997).

**MLPs directly trained on human choices.** We also implemented Multilayer Perceptrons (MLPs) with a single hidden layer containing 320 neurons and using the sigmoid activation function. These MLPs were directly trained on each of the four human datasets, using the stimuli features as input to predict choice rates. These MLPs potentially capture an upper bound of the explainable variance within human data (Agrawal et al., 2020). However, they may have overlooked significant constraints from the original psychological experiments, as these models use task features as input rather than the actual stimuli presented in texts or the stylized representations required for rational agents, as in our Arithmetic-GPT.

**Choice probabilities and embeddings from open-weight LLMs.** To further investigate the impact of different input formats and compare with off-the-shelf LLMs, we evaluated the performance of LLaMA-3-70B-Instruct on human choice data. Unlike Arithmetic-GPT, the LLaMA3 model not only excels in arithmetic tasks but also comprehends and generates human-like text. This capability results from its training on extensive text data and a significantly larger number of model parameters. In short, LLaMA3 is a more versatile and powerful model, also based on the transformer architecture and trained autoregressively.

Given these features of LLaMA3, we presented each choice problem in two different formats: text-based and arithmetic equation-based. The text format converts each choice problem into a descriptive narrative, simulating the stimuli presented to human participants (see Appendix A for a detailed description of the text prompts). We instructed the model to report its selection between the two options, using the log probability of the chosen option to determine the model's predicted choice rates for the corresponding option. In contrast, the arithmetic-equation format presents the choice problems as a series of arithmetic computations required by a rational agent. Note that this format is identical to the input used for Arithmetic-GPT. We obtained the embeddings from LLaMA3 for each choice problem represented in the arithmetic-equation format.

---

[2]https://llama.meta.com/llama3/

Table 3: Proportion of the variance of human choices explained ($R^2$) by embeddings from the pretrained Arithmetic-GPT model compared to other computational models. Bold numbers indicate the best models within each group.

| Model | Training data | Risky choices | | Intertemporal choices | |
|---|---|---|---|---|---|
| | | `choices13k` | `cpc18` | `gershman20` | `agrawal23` |
| Arith.-GPT | Synthetic (unif.)[a] | 69.3% | 63.2% | 64.0% | **96.1%** |
| Arith.-GPT | Synthetic (unif. abl.)[b] | 57.5% | 37.9% | 59.8% | 81.1% |
| Arith.-GPT | Synthetic (eco.)[c] | **70.8%** | **65.5%** | **67.8%** | 95.5% |
| Arith.-GPT | Synthetic (eco. abl.)[d] | 61.4% | 33.8% | 60.7% | 80.1% |
| Arith.-GPT | None[e] | 21.0% | 28.4% | 10.8% | 21.1% |
| LLaMA3 (txt.)[f] | Undisclosed[g] | 14.2% | 8.3% | 4.0% | 3.0% |
| LLaMA3 (txt emb.)[h] | Undisclosed[g] | **66.2%** | **55.7%** | 66.5% | **98.0%** |
| LLaMA3 (arith.)[i] | Undisclosed[g] | 63.6% | 34.8% | **69.3%** | 96.0% |
| Prospect theory | Human choices | 51.5% | 54.2% | N/A | N/A |
| Hyperbolic[j] | Human choices | N/A | N/A | 53.4% | 36.1% |
| Expected value[k] | None | 31.6% | 43.9% | 41.7% | 26.1% |
| MLP | Human choices | **82.7%** | **97.8%** | **60.6%** | **94.0%** |

*Note.* [a]Uniform synthetic datasets. [b]The same uniform synthetic datasets but with the answers on the right-hand side of the equations removed and the signs randomized between positive and negative. [c]Ecological synthetic datasets. [d]The same ecological synthetic datasets but with the answers on the right-hand side of the equations removed and the signs randomized between positive and negative. [e]Embeddings from an untrained Arithmetic-GPT model with randomly initialized weights. [f]Log probabilities of LLaMA3 elicited from text descriptions of choice problems. [g]The training data is not publicly available, but Meta has disclosed some summary statistics of the training corpus. [h]Embeddings of LLaMA3 elicited from text descriptions of choice problems. [i]Embeddings of LLaMA3 elicited from arithmetic equations of choice problems. [j]The hyperbolic discounting model for intertemporal choices. [k]Logistic regression results when applied to the expected value difference between the two choice options.

## 5 EXPERIMENTAL RESULTS

### 5.1 MODEL COMPARISONS

In this section, we present the experimental results from our model comparisons. We first obtained embeddings from Arithmetic-GPT and LLaMA3 models, evaluating versions of the model that were pretrained on each of our four distinct synthetic datasets as well as a version without any training. Specifically, we extracted embeddings for the expected values of the two options, denoted as $e_A$ and $e_B$. Additionally, we obtained embeddings for the difference in expected values, denoted as $e_{A-B}$. All embeddings were derived from the representation in the final layer before the autoregressive prediction. We then performed a logistic regression using $e_A$, $e_B$, and $e_{A-B}$ as independent variables, with human choice probabilities as the dependent variable. Adjusted $R^2$ values were used for all logistic regressions, including those for Arithmetic-GPT, LLaMA3, and EV results. All other $R^2$ values were reported as the squared Pearson's $r$ between model predictions and human data. These results indicate that the embeddings from the Arithmetic-GPT model pretrained on ecologically valid synthetic datasets most accurately capture human choices. This model also outperforms the embeddings obtained from the LLaMA3 model (the 7th row of Table 3), suggesting that pretraining on synthetic datasets is sufficient to create a strong correspondence between LLMs and human decision-making. However, the LLaMA3 model performs comparably to Arithmetic-GPT in predicting intertemporal choices. The log probabilities from the same LLaMA3 model perform poorly in comparison to human data (the 6th row of Table 3), replicating previous findings using choice rates reported from LLaMA1 (Binz & Schulz, 2023a). The $R^2$ values between Arithmetic-GPT pretrained on uniform and ecological synthetic datasets are small yet consistent. This may be due to the limited range tested in the human datasets.

To benchmark the performance of Arithmetic-GPT models in explaining human data, we directly fit behavioral models and MLPs on each of the human choice datasets. Prospect theory and the

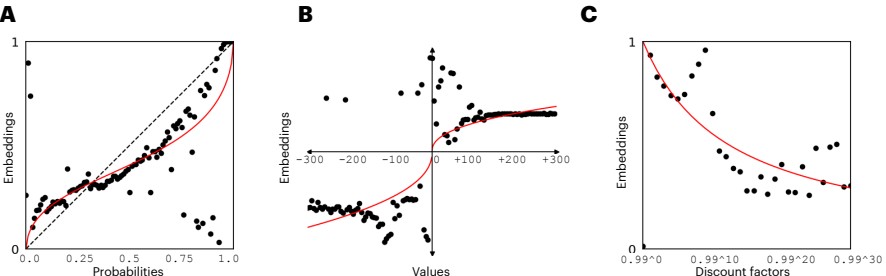

Figure 2: Embeddings from ecologically pretrained Arithmetic-GPT for inputs including **(A)** probabilities, **(B)** values, and **(C)** discount factors. Inputs are shown along the horizontal axes and embeddings are shown on the vertical axes. The embeddings, shown as black dots, were reduced to 1D using multidimensional scaling. Embeddings for probabilities and discount factors are normalized between 0 and 1 (see Appendix D for details). The red curves represent the best-fitting behavioral economic models: **(A)** the probability weighting function from PT with best-fitting $\gamma = 0.58$, **(B)** the utility function from PT with best-fitting $\alpha = 0.42, \beta = 0.45, \lambda = 1.4$, and **(C)** the hyperbolic discount function with best-fitting $k = 0.08$.

hyperbolic discounting model are leading behavioral models in risky and intertemporal choices, respectively (Kahneman & Tversky, 2013; Laibson, 1997). The behavioral models have interpretable mechanisms and contain few free parameters. However, they do not explain human data as well as the embeddings from both LLaMA3 and Arithmetic-GPT, although they still outperform a simple choice model based on EV difference. Moreover, prospect theory and the hyperbolic discounting model do not generalize to both risky and intertemporal choices, resulting in N/A values in their respective rows of Table 3.

In contrast, fitting an MLP directly on human data potentially reveals the ceiling performance of any model in explaining the data (Agrawal et al., 2020; Peterson et al., 2021). Except for the intertemporal choice tasks, MLPs outperform all other models. It is important to note that MLP training was based on task features rather than text descriptions that simulated participants' experiences or arithmetic equations that mimic a rational agent's computations. Consequently, these differences in input formats could also lead to diverging performance. The arithmetic format of intertemporal choice may provide a better fit for human data. Indeed, there is increasing evidence from experimental economics suggesting that the complexity of discounting values, even in the absence of actual payoff delays, influences intertemporal choices (Enke et al., 2023). We include a robustness check using 10-fold cross-validation in Appendix B.

## 5.2 IMPLICIT FUNCTIONS OF PROBABILITIES, VALUES, AND DISCOUNT FACTORS

To understand why Arithmetic-GPT, pretrained to calculate expected values with ecological distributions of probabilities and values, can capture human choices, we examined the implicit functions of probabilities, values, and times derived from the model embeddings. We extracted embeddings for probabilities ranging from $0.0$ to $1.0$, values ranging from $-300$ to $+300$, and discount factors ranging from $0.99^0$ to $0.99^{30}$. These high-dimensional embeddings were then reduced to a single dimension using multidimensional scaling with Euclidean distance. Additionally, for probabilities and discount factors, we normalized the embeddings to a range between 0 and 1.

We find that the embeddings from ecologically pretrained Arithmetic-GPT replicate classical findings from behavioral economics, including value and probability weighting functions from the prospect theory (Tversky & Kahneman, 1992) and the hyperbolic discounting function (Laibson, 1997). Specifically, probabilities close to 0.5 are more similar to each other than to probabilities close to either 0 or 1 (see Figure 2A and Equation 4). Embeddings for values illustrate concavity for positive values, convexity for negative values, and a steeper slope for negative values than for positive values (see Figure 2B and Equation 3). These features reflect risk aversion for gains, risk seeking for losses, and loss aversion (Kahneman & Tversky, 2013; Tversky & Kahneman, 1992). Moreover, embeddings for the discount factor demonstrate that distant times are more similar (see

Figure 2C and Equation 6), enabling a present-bias (Meier & Sprenger, 2010). However, embeddings from Arithmetic-GPT pretrained on non-ecological datasets (i.e., ablated ecological, uniform, and ablated uniform) and those from LLaMA3 exhibit fewer human-like distortions in their implicit functions (see Appendix E for details).

For the probability weighting function, the curvature parameter ($\gamma$) of human participants was found to be 0.61 for gains and 0.69 for losses (Tversky & Kahneman, 1992). Follow-up replications by (Wu & Gonzalez, 1996) found $\gamma$ values around 0.7. Moreover, the utility curvature parameters ($\alpha$ and $\beta$) typically range between 0.5 and 0.9, while the loss aversion parameter ($\lambda$) is approximately 2.25 (Tversky & Kahneman, 1992). Regarding human intertemporal choices, typical $k$ values range between 0.01 and 0.1 for small magnitude studies (Odum, 2011). Comparing to these best-fitting parameters from human experiments, we observe that the implicit functions derived from the embeddings of Arithmetic-GPT also quantitatively match those observed in humans (see Figure 2). The implicit functions, however, exhibit discontinuities, suggesting that the smooth functions derived from human theories may not be directly applicable to explain the embeddings of Arithmetic-GPT.

## 6 DISCUSSION

We introduced an approach to transforming language models into cognitive models by pretraining them on tasks that mirror the computational process performed by a rational agent. Specifically, we pretrained Arithmetic-GPT to calculate expected values using ecologically distributed probabilities and values. This pretraining allows the model to better capture human risky and intertemporal choices compared to classical behavioral models and, in some cases, even surpass the performance of the general-purpose LLaMA3 model. These results suggest that the observed cognitive biases in LLMs may stem from the training task, architecture, and numerical reasoning abilities, without requiring natural language supervision. These results have implications for a number of questions in cognitive science and machine learning, although we also note the limitations of our current work.

**Language Models as Cognitive Models.** There is a growing research effort focused on exploring LLMs as scientific tools for understanding the human mind (Binz & Schulz, 2023a; Frank, 2023b; Horton, 2023). Despite the challenges outlined in Section 2, LLMs offer unique opportunities for investigating cognitive processes in ways that are not feasible with human participants. Recent studies have even demonstrated that fine-tuning a LLaMA1 model on human data allows the model to outperform classical models in explaining this data (Binz & Schulz, 2023a). While the LLaMA series' model architecture and weights are publicly available, researchers lack access to the training data, which makes crucial scientific inference practically impossible. In contrast, we explicitly manipulate the training data for our Arithmetic-GPT, thereby uncovering a key factor that contributes to the model's ability to explain human choices. This targeted approach, however, comes at the cost of the broader versatility inherent in off-the-shelf LLMs, which are designed to handle a wider range of tasks. Notably, language models, including Arithmetic-GPT, do not directly model human cognitive processes. Instead, our work suggests that these models are better suited for probing the computational problems that human cognition aims to solve.

**Bayesian Models of Cognition, Meta-learning, and Pre-training.** Bayesian models of cognition have been instrumental in understanding human performance across a variety of cognitive tasks by providing optimal solutions to the inductive inference problem. Recently, it has been argued that Bayesian models and neural network models can be viewed as complementary to one another (Griffiths et al., 2023). Neural networks that are meta-trained have also been shown to exhibit properties consistent with Bayesian models (Lake & Baroni, 2023; McCoy & Griffiths, 2023). Moreover, pre-training a neural network model for improved performance on downstream tasks can be seen as a form of meta-learning or the acquisition of useful inductive biases, similar to Bayesian models (Hsu et al., 2018). Our work makes the implicit priors learned by Arithmetic-GPT more explicit by specifying the synthetic dataset on which it was pre-trained.

**Computationally Equivalent Tasks for Cognitive Modeling.** Modeling human cognition is challenging because the hypothesis space of possible cognitive mechanisms that can explain human data equally well is vast. This makes principles of rationality desirable, as assuming people are rational in some sense greatly constrains the hypothesis space, thereby making scientific inference more effective. However, human rationality has been a subject of debate in economics and psychology for over a century (von Neumann & Morgenstern, 1944; Kahneman & Tversky, 2013; Simon, 1997). While

significant progress has been made in understanding human rationality (e.g., Lieder & Griffiths, 2020; Gershman et al., 2015), the advent of LLMs seems to challenge the need for rational theories. Simply training LLMs to predict the next word appears sufficient to produce human-like behaviors, suggesting that we can model human cognition without the constraints imposed by rational theories. However, our experimental results suggest an alternative route to enhancing the correspondence between behaviors produced by LLMs and humans: pretraining on computationally equivalent tasks that a rational agent would need to master. Future research should investigate the impact of different assumptions about the nature of rationality on task content and distributions, and explore whether there are more effective assumptions for pre-training models to explain human behavior.

**Implications for Theories of Human Risk and Time Preferences.** The success of Arithmetic-GPT in explaining human risky and intertemporal choices has significant implications for theoretical work on human risk and time preferences. While the *Homo economicus* portrayal of human beings as perfectly rational and self-interested agents has been inadequate in describing human choices (Gigerenzer & Gaissmaier, 2011; Kahneman & Tversky, 2013), existing behavioral models that deviate from rationality do not generalize well across different task domains. For instance, it is challenging to use prospect theory to model time preferences or to use the hyperbolic discounting model to explain risk preferences. In contrast, Arithmetic-GPT has demonstrated substantial transferability across task domains. The same model, in principle, can be adapted to other judgment and decision-making tasks such as social choice, probability judgment, and belief updating. The key factor enabling language models to generalize across a wide range of tasks is the presence of a language interface, which underpins a significant range of cognitive tasks.

**The Importance of Training Data Disclosure.** Our results demonstrate that the training data used for LLMs is crucial for understanding their emergent capabilities and the inductive biases they acquire during pretraining. Adjusting the distribution of, or ablating, the training data can significantly affect the degree to which LLMs correspond with human behaviors. These findings suggest that existing off-the-shelf LLMs, whether proprietary or open-weight, including the GPT series (Brown et al., 2020) and the LLaMA series (Touvron et al., 2023), are less effective as models of human cognition. This is primarily because their training data is rarely disclosed, making it difficult for scientists to control for data contamination and thereby precisely identify the sources of human-like behaviors in these models.

**Limitations and Future Research.** While we have made progress in addressing many challenges associated with using language models as cognitive models, some issues remain unresolved. To address the data gap between LLMs and human learners, we limited the scope of our synthetic dataset to the arithmetic of expected value calculations. Despite this, LLMs still require a substantial amount of training data to perform arithmetic accurately within a limited range of input values. Moreover, it is unrealistic for human learners to process 1M randomly generated mathematical equations, as Arithmetic-GPT did, to acquire the skill of computing expected values. Further research is needed to continue bridging this data gap (but see Appendix F for an initial attempt).

Additional ablation studies could be performed on model architectures and training objectives. Our work is fundamentally limited to autoregressive training and a decoder-only transformer architecture. We believe that alternative training mechanisms and model architectures could potentially yield better embeddings from language models. One area for future research is estimating the lower bounds on model size necessary to achieve a certain level of correspondence with human behavior. Another area of future work involves leveraging interpretability techniques to distill novel cognitive mechanisms from Arithmetic-GPT.

**Conclusion.** Large language models have opened new horizons for research on human cognition, but also introduce a new set of challenges based on the volume of training data, the content of those data, the influence of value alignment, and limited access to the training regimes and weights of these models. We have proposed an approach to addressing these challenges, based on training small language models with datasets that are generated based on tasks that are hypothesized to be computationally related to the problem that human minds face. Our results show that this approach is extremely effective in predicting human decisions, where training on arithmetic results in representations that can be used to predict human choices better than both existing psychological models and large language models trained on broader datasets. This approach is easily generalizable to other cognitive tasks that primarily rely on language interface, and can even be used with other kinds of foundation models to study human perception.

**Acknowledgments.** This work and related results were made possible with the support of the NOMIS Foundation. H. Yan acknowledges the Chancellor's International Scholarship from the University of Warwick for additional support.

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

# A  PROMPTS

### EXAMPLE OF A RISKY CHOICE DESCRIBED IN TEXT

**System:**  `You are participating in a gambling game where you will be shown two options, Gamble A and Gamble B. Your task is to choose between the two.  You must select one option.`

**User:**  `Gamble A offers a 10% chance to win $10 and a 90% chance to lose $12.  Gamble B offers a 40% chance to lose $13 and a 60% chance to win $22.  Please limit your answer to either 'A' or 'B'.`

### EXAMPLE OF AN INTERTEMPORAL CHOICE DESCRIBED IN TEXT

**System:** `You are participating in an intertemporal choice experiment. You will be presented with two options, A and B, and your task is to choose between them.  You must select one option.`

**User:**  `Option A offers $80 after 30 days.  Option B offers $13 immediately.  Please limit your answer to either 'A' or 'B'.`

# B  ROBUSTNESS CHECK

We supplement our main results with a robustness check using 10-fold cross-validation. Each dataset of human choices was randomly partitioned into a 90% training set and a 10% validation set. For the `gershman20` dataset, a stratified train/validation split was employed due to the certainty equivalence design of the experiment (Gershman & Bhui, 2020), ensuring that choices made within the same problem were grouped together.

Embeddings from Arithmetic-GPT models and LLaMA-3-70B-Instruct were fitted using logistic regression with a LASSO penalty on the training set. MLPs were fitted to task features in the training set. Performance for all models was assessed on the validation set, with $R^2$ calculated as the squared Pearson's $r$. This process was repeated 10 times, each with a random train/validation split. The mean and standard error (SE) of $R^2$ are summarized in Table A1, demonstrating a replication of our main results presented in Table 3. Finally, we conducted a model comparison using the entire `choices13k` dataset, including the decision-from-feedback trials (see Table A2). We observe that LLaMA3 can outperform MLP on the `agrawal23` dataset. While the two neural network models use different task formats as inputs, which may contribute to the differences in performance, it is also possible that the dataset lacks sufficient statistical power to reliably distinguish between the two models.

Table A1: Cross-validation results. Propotion of the variance in human choices explained ($R^2$) on the validation set for each model. Numbers in parentheses represent standard errors from 10-fold cross-validation.

| Model | Training data | Risky choices | | Intertemporal choices | |
|---|---|---|---|---|---|
| | | `choices13k` | `cpc18` | `gershman20` | `agrawal23` |
| Arith.-GPT | Synthetic (unif.) | 56.4% (0.4%) | **46.3%** (3.1%) | **56.7%** (1.1%) | 81.5% (0.4%) |
| Arith.-GPT | Synthetic (unif. abl.) | 38.5% (1.5%) | 21.9% (4.6%) | 51.1% (0.7%) | 54.0% (0.3%) |
| Arith.-GPT | Synthetic (eco.) | **57.7%** (0.4%) | **51.7%** (4.9%) | **57.1%** (1.2%) | **83.7%** (0.3%) |
| Arith.-GPT | Synthetic (eco. abl.) | 38.5% (1.5%) | 33.6% (5.0%) | **55.3%** (1.2%) | 68.3% (0.2%) |
| Arith.-GPT | None | 33.1% (1.5%) | 23.5% (3.8%) | 15.3% (1.7%) | 50.4% (5.1%) |
| LLaMA3 (txt emb.) | Undisclosed | 54.0% (0.8%) | 35.0% (5.0%) | 52.7% (1.1%) | 89.9% (0.1%) |
| LLaMA3 (arith.) | Undisclosed | 38.6% (1.5%) | 20.8% (3.0%) | 54.9% (1.2%) | 87.6% (0.1%) |
| MLP | Human choices | 62.3% (1.2%) | 50.3% (3.9%) | 58.7% (1.3%) | 84.9% (2.4%) |

*Note.* The bold numbers represent the top-performing variants of the Arithmetic-GPT models, as measured by $R^2$. In cases where multiple models are highlighted, the differences in $R^2$ values are not statistically significant at the $p = 0.05$ level.

Table A2: Proportion of the variance in human choices explained ($R^2$) on the entire `choices13k` dataset for each model. Bold numbers indicate the best models within each group.

| Model | Training data | Entire `choices13k` |
|---|---|---|
| Arith.-GPT | Synthetic (unif.) | 64.0% |
| Arith.-GPT | Synthetic (unif. abl.) | 49.2% |
| Arith.-GPT | Synthetic (eco.) | **64.3%** |
| Arith.-GPT | Synthetic (eco. abl.) | 52.1% |
| LLaMA3 (txt.) | Undisclosed | 14.6% |
| LLaMA3 (txt emb.) | Undisclosed | **65.6%** |
| LLaMA3 (arith.) | Undisclosed | 60.6% |
| Prospect theory | Human choices | 45.6% |
| Expected value | None | 40.2% |
| MLP | Human choices | **73.8%** |

## C    LEARNING CURVE

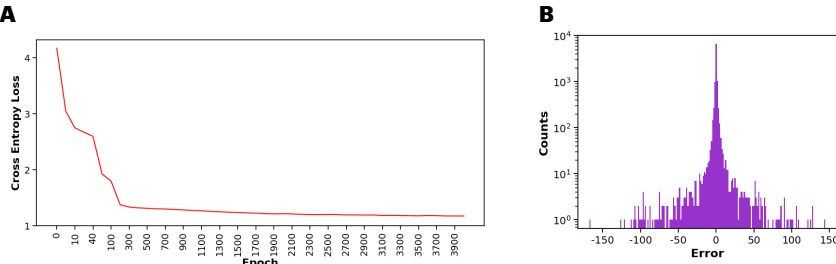

Figure A1:    **(A)** Training loss decreases over the course of training epochs for Arithmetic-GPT trained on ecological synthetic dataset. **(B)** Histogram displaying the differences between the top-1 responses of the pretrained Arithmetic-GPT model and the actual expected values in the ecological synthetic dataset.

## D    DIMENSIONALITY REDUCTION DETAILS

We used multidimensional scaling to reduce the dimensionality of embeddings to 1D, with Euclidean distance serving as the dissimilarity metric for forming the 1D manifolds. Embeddings of probabilities and discount factors were normalized using max-min normalization to ensure the resultant values fell within the range of $[0, 1]$.

## E    ADDITIONAL IMPLICIT FUNCTIONS

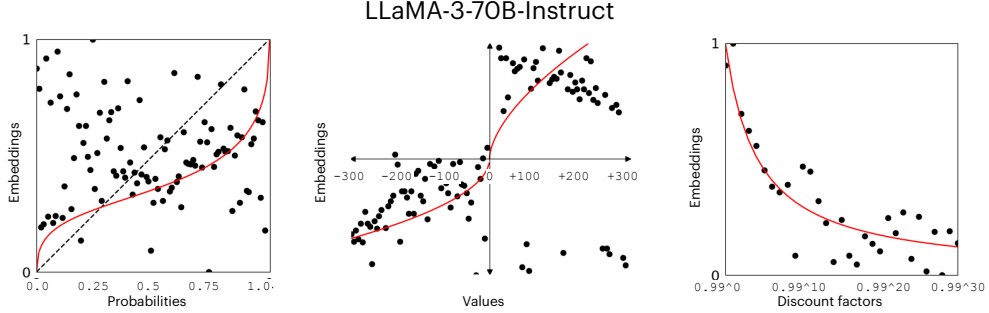

Figure A2:   Visualizations of embeddings from LLaMA-3-70B-Instruct.

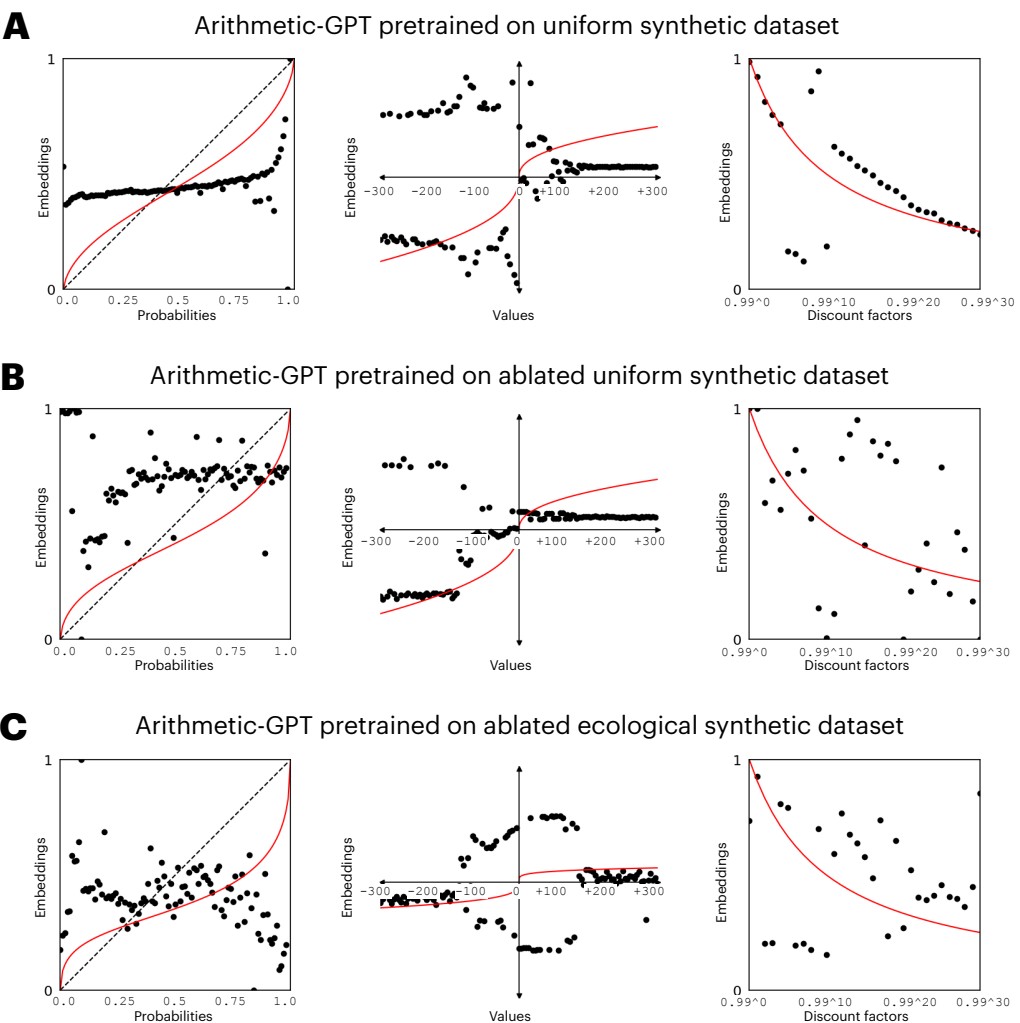

Figure A3: Visualizations of embeddings from Arithmetic-GPTs pretrained on different synthetic datasets. **(A)** Uniform synthetic dataset. **(B)** Ablated uniform synthetic dataset. **(C)** Ablated ecological synthetic dataset.

## F  VARIATIONS IN MODEL SIZES, DATA QUANTITY, AND DATA DISTRIBUTIONS

In this section, we explore how adjustments to key hyperparameters of Arithmetic-GPT impact the quality of its pretrained embeddings. Specifically, we investigate the effects of reducing the model's hidden size from 320 (as reported in the main text) to 104 and 16. Moreover, we examine the influence of decreasing the size of the ecological synthetic dataset, reducing it from 1M mathematical equations (as reported in the main text) to subsets of 100K and 10K equations. Apart from these modifications, we adhered to the same pretraining procedure and evaluated the resulting embeddings on the two largest datasets, `choices13k` for risky choices (Peterson et al., 2021) and `agrawal23` (Agrawal et al., 2023) for intertemporal choices, using 10-fold cross-validation.

The results, summarized in Table A3, reveal an intriguing pattern: reducing the size of the ecological synthetic dataset to as few as 10K equations does not significantly impair the ability of the 10M model's embeddings to predict human choices. It is important to note that Arithmetic-GPT models were never exposed to human data during pretraining. However, significant reductions in the model

size – particularly 30K parameters – lead to a decline in performance, highlighting the critical role of model capacity in embedding quality.

Table A3: Variants of Arithmetic-GPT models pretrained on ecological synthetic datasets of varying sizes. In each cell, $R^2$ results (in percentage) are reported as `choices13k` (left of '/') and `agrawal23` (right of '/'). Numbers in parentheses indicate standard errors obtained from 10-fold cross-validation.

| | | Model Sizes | | |
| --- | --- | --- | --- | --- |
| | | 10M (hidden size=320) | 1M (hidden size=104) | 30K (hidden size=16) |
| | 1M | **57.7** (0.4) / **83.7** (0.3) | 54.7 (0.6) / **83.4** (0.3) | 30.6 (0.4) / 56.8 (0.4) |
| Data quantity | 100K | **57.2** (0.6) / **83.6** (0.4) | 55.5 (0.5) / 81.6 (0.4) | 31.4 (0.8) / 63.4 (0.2) |
| | 10K | **57.7** (0.5) / **83.6** (0.3) | 50.0 (0.5) / **84.1** (0.3) | 29.1 (0.6) / 63.0 (0.4) |

*Note.* The bold numbers represent the top-performing variants of the Arithmetic-GPT models, as measured by $R^2$. In cases where multiple models are highlighted, the differences in $R^2$ values are not statistically significant at the $p = 0.05$ level.

We conducted an exploratory analysis using the original Arithmetic-GPT model architecture and synthetic datasets of 1M equations to investigate the effects of varying data distributions. Specifically, we independently manipulated the distributions of probabilities and values. For probability distributions, we considered Beta(0.27, 0.27) (ecological), Beta(1, 1) (uniform), and Beta(2, 2). For value distributions, we examined power-law distributions with exponents of 0 (uniform), -0.945 (ecological), and -2. All Arithmetic-GPT models were randomly initialized and trained following the procedure outlined in Section 3.1.

As shown in Table A4, aligning probability distributions with ecological patterns, such as Beta(0.27, 0.27), generally enhances the quality of embeddings for predicting risky choices. Conversely, adopting a power-law distribution for values with an exponent of -2 tends to improve the model's performance in predicting intertemporal choices.

Table A4: Variants of Arithmetic-GPT models pretrained on different data distributions. In each cell, $R^2$ results (in percentage) are reported as `choices13k` (left of '/') and `agrawal23` (right of '/'). Numbers in parentheses indicate standard errors obtained from 10-fold cross-validation.

| | | Distribution of probabilities | | |
| --- | --- | --- | --- | --- |
| | | Beta(0.27,0.27) | Beta(1,1) | Beta(2,2) |
| | 0 | 56.5 (0.5) / 82.5 (0.4) | 56.4 (0.4) / 81.5 (0.4) | 56.0 (0.3) / 77.5 (0.4) |
| Distribution of values (exponent of power-law) | -0.945 | **57.7** (0.4) / 83.7 (0.3) | 56.7 (0.3) / 78.4 (0.5) | 56.5 (0.3) / 77.6 (0.4) |
| | -2 | 55.7 (0.3) / **85.9** (0.3) | 56.3 (0.5) / 82.6 (0.4) | 55.5 (0.4) / 82.2 (0.4) |

*Note.* The bold numbers represent the top-performing variants of the Arithmetic-GPT models, as measured by $R^2$.

## G    COMPUTATIONAL ABILITIES OF LANGUAGE MODELS

We evaluate the abilities of LLaMA-3-70B-Instruct and Arithmetic-GPT in computing expected values using two newly generated test datasets. Both test sets consist of 20K synthetically generated equations involving expected value calculations. In one test set, the probabilities and values in the equations are sampled from a uniform distribution, while the other test set features probabilities and values derived from ecological distributions, as illustrated in Figure 1. The train-test relationships for these datasets are summarized in Table A5.

The computational ability of Arithmetic-GPT models was assessed by calculating the probability of generating correct expected values (rounded to two decimal places). Formally, the correct probability is computed as $P_{\text{model}}$(true expected values | the left-hand side of the equation), using token probabilities. Arithmetic-GPT, when pretrained on ecological distributions, demonstrated higher

probabilities of generating correct values across both uniform and ecological test sets. When comparing performance on the same model across the two test sets, expected values in the ecological test set were found to be easier to predict. One possible explanation for the improved performance of Arithmetic-GPT models pretrained on ecological distributions is that these distributions are more "bursty," characterized by multiple occurrences of similar numerical values (cf. Chan et al., 2022). Pretraining on bursty sequences has been shown to facilitate emergent behaviors in LLMs, such as in-context learning (Chan et al., 2022).

To evaluate LLaMA3's ability to compute expected values, we directly prompted the model to solve arithmetic equations while setting the temperature to 0. The integer part of the expected values was extracted from LLaMA3's responses by filtering out irrelevant tokens. The correct probability was calculated as the relative frequency of correctly predicting the integer part of the true expected values across 20K questions. As shown in Table A5, LLaMA3 performed poorly compared to Arithmetic-GPT, achieving only 20.97% and 38.79% accuracy on the uniform and ecological test sets, respectively.

The model's behavioral performance on the ecological test set broadly predicts the ranking of its embeddings in modeling human data. However, directly interpreting a language model's performance on arithmetic tasks in relation to model-fitting results that leverage its embeddings can be challenging. Therefore, further investigation is necessary to establish a robust connection between a language model's arithmetic capabilities and the predictive power of its embeddings in predicting human choices.

Table A5: Evaluation of language models' probabilities of generating correct expected values across different train-test distributions. Numbers in parentheses represent standard errors.

| Model | Training data | Test data | Correct probability |
|---|---|---|---|
| Arith.-GPT | Synthetic (unif.) | Uniform | 0.8261 (0.0023) |
| | | Ecological | 0.9336 (0.0014) |
| Arith.-GPT | Synthetic (eco.) | Uniform | 0.9388 (0.0013) |
| | | Ecological | 0.9780 (0.0007) |
| LLaMA3 | Undisclosed | Uniform | 0.2097 |
| | | Ecological | 0.3879 |

## H    IMPLEMENTATION DETAILS

Here we detail the hyperparameter setup for experiments. All computations for synthetic datasets are run on single Nvidia RTX 3060 GPU, and those for LLaMA3 embeddings are run on single A100 GPU.

