# OpenReview forum: "Language Models Trained to do Arithmetic Predict Human Risky and Intertemporal Choice"
_ICLR.cc/2025/Conference — ICLR 2025 Poster_

### Official Review · Reviewer_27fR · 2024-10-27

**Soundness:** 3
**Presentation:** 4
**Contribution:** 3
**Rating:** 8
**Confidence:** 4

**Summary:**

The authors explore how language models can serve as cognitive models by investigating their behavior in risky and intertemporal choice tasks. The authors introduce Arithmetic-GPT, a small language model specifically trained on arithmetic tasks that simulate expected value (EV) and present value (PV) calculations essential to decision-making. By training Arithmetic-GPT (A-GPT) on a non-ecologically and ecologically valid datasets, the authors show that A-GPT exhibits decision-making patterns that closely resemble human choices (in particular for the ecologically valid dataset). Moreover, A-GPT outperforms both traditional behavioral models and some large, general-purpose language models in predicting human risk and time preferences. The work porposed by the authors underscores the importance of tailored training datasets for aligning language model behavior with human cognitive processes and provides insights into how synthetic datasets may enhance models' ability to predict human decision patterns.

**Strengths:**

1. This work exemplifies the potential of LLMs to gain insights in human cognitive processing, and as a results showcases the elegance of the auhtors thought process.

2. The auhors compare their model to a relevant suite of other models, and use several datasets to drive their point. This systematic comparison highlights the quality of this work and further amplifies the results obtained with A-GPT.

3. The paper is clear and well structured, allowing for smooth reading. The figures are informative and clearly illustrate the points made in the text.

**Weaknesses:**

Within the range of what the paper is tackling, I do not see any weaknesses in this work.

I do however believe that the authors could have gained further understanding of human risky and intertemporal choices by generating several other datasets that systematically target specific aspect of EV and PV calculations, i.e., parametrically changing the ablation level of specific EV and PV compononents.

Moreover, it would be interesting to relate the A-GPT results with explainability in order to potentially provide more interpretability to the embeddings.

I will however not discount any points in my rating. I understand the limitations in terms of computation,cost, time and manuscript space.

**Questions:**

Typo:

There is an issue with the presentation of Figure A2 which falls under appendix F rather than E.

---

> ### Author Response · Authors · 2024-11-22
>
> Thank you for the positive evaluation of our work.
>
> **No weakness within the scope of the work. But it would be nice to see some parameterized ablation level for synthetic datasets.**
>
> **Response:** Thank you for this suggestion. Indeed, the potential distributions of probabilities and values that could be tested are virtually limitless. As an initial step, we explored three probability distributions by varying the Beta distribution parameters and three value distributions by adjusting the exponent of a power-law. Detailed results are provided in Table A4 of the revised manuscript.
>
> In summary, our findings indicate that aligning probability distributions with ecological patterns improves the model’s ability to predict risky choices. Conversely, shaping value distributions as a power-law with an exponent of -2 enhances the model’s performance in predicting intertemporal choices.
>
> **Relate to interpretability research to better understand Arithmetic-GPT’s embeddings.**
>
> **Response:** We agree that applying modern interpretability techniques to Arithmetic-GPT could offer valuable insights into what cognitive mechanisms it has encoded. As noted in the “Limitations and Future Research” section, interpretability research will be a key focus in our future work.
>
> **Presentation issue in Fig A2**
>
> **Response:** Thank you! It is fixed now with Fig A2 under Appendix E rather than F.

---

### Official Review · Reviewer_eVJL · 2024-11-02

**Soundness:** 3
**Presentation:** 3
**Contribution:** 3
**Rating:** 6
**Confidence:** 3

**Summary:**

Humans often deviate from optimal choices in tasks involving expected value or temporal patterns. This behavior is also observed in LLMs, though embeddings do not fully capture this deviation. This study explores how this tendency arises in language models by pre-training a small model (Arithmetic-GPT) on expectation calculation tasks then testing its alignment with human choices.

Synthetic datasets (uniform and ecological, each with perturbed and unperturbed versions) are created for model pre-training and tested on four human datasets related to risky and inter-temporal decisions. Model comparisons included pre-trained Llama (natural language input + log-likelihoods, natural language + text embeddings, arithmetic + embeddings), Arithmetic-GPT variants (four dataset versions and no training), four behavioral models, and MLPs directly trained on task features. Results show that Arithmetic-GPT trained on unperturbed ecological data shows the highest explained variance with human decisions (input: embeddings/log-likelihoods, output: human decisions).

**Strengths:**

- **Originality**:
     - Proposes a new test environment to evaluate the effectiveness of using language models as cognitive models.
     - The ecological data pre-training approach is a simple approach to model principles with human decision-making patterns.

 - **Quality and Clarity**:
     - Comprehensive benchmarking across multiple models, including classical behavioral and neural models.
     - Very clear writing, but with minor typos (e.g., line 376, "fitted" → "fit").

**Weaknesses:**

Overall, I am uncertain if the proposed task is too simple to make any sophisticated claims on language models as cognitive models.

 - Does Arithmetic-GPT’s performance really represent actual human decision processes, or is it not possible that Arithmetic-GPT's "human-like behavior” could instead be a reflection of general statistical tendencies that happen to deviate from optimal choice, rather than an insight into genuine human cognition?
- Table 3 shows that Arithmetic-GPT only marginally outperforms the larger Llama model. This suggests that Llama, without specialized pre-training, already captures much of the variance in human choices. I suppose the claim is somewhat valid given the limited capacity of the Arithimetic-GPT backbone, but the authors’ emphasis on Arithmetic-GPT's superiority might be overstated given that its improvement is slight and comes from training on targeted synthetic data.

**Questions:**

- A minor question regarding the experiments: Why not also try training Llama or Arithmetic-GPT on the task features as the MLP?

---

> ### Author Response · Authors · 2024-11-22
>
> We thank the reviewer for the constructive comments and feedback.
>
> **Weakness 0: The risky and intertemporal tasks are too simple to support sophisticated claims about language models as cognitive models.**
>
> **Response:** Risky and intertemporal choices are fundamental forms of decision-making as they reflect an agent’s risk and time preferences, which are core constructs in cognitive psychology and behavioral economics for studying decision-making. These tasks are considered the gold standard for studying human decisions, inspiring hundreds of papers in psychology and economics, and serve as essential building blocks for more complex cognitive tasks. Therefore, it is critical for any cognitive model to perform well in these domains. Moreover, two of the human datasets used in our study are large-scale experiments in the risky and intertemporal choice domains, providing robust statistical power for modeling human behavior.
>
> **Weakness 1: Should we view the Arithmetic-GPT as a process model that hypothesizes actual cognitive processes?**
>
> **Response:** We do not view Arithmetic-GPT as a model of the actual cognitive processes behind human decision-making. Instead, we use language models as tools to explore computational principles underlying human decisions. The key advantage of language models here is their ability to encode symbols (e.g., values, probabilities, and discount factors) as vectors (e.g., model embeddings) shaped by the training task. Our approach reveals that ecologically training the model on expectation calculations creates human-like representations. Overall, our method aims to uncover computational principles in human cognition (e.g., how human-like behavior emerges in specific cognitive tasks as a consequence of solving what kind of computational challenges), without assuming specific cognitive processes that implement these computations. To clarify this we added the following text in our Discussion section:
> “Notably, language models, including Arithmetic-GPT, do not directly model human cognitive processes. Instead, our work suggests that these models are better suited for probing the computational problems that human cognition aims to solve. ”
>
> **Weakness 2: Arithmetic-GPT is marginally better than LLaMA. LLaMA has already captured much of the variance in human choices.**
>
> **Response:** We agree that LLaMA-3-70B-Instruct captures a significant portion of the variance in human data. However, as noted in the Background section, it is challenging to pinpoint why LLaMA performs well in explaining human data due to potential data contamination. LLaMA may have been trained on human data or exposed to literature on human biases in risky and intertemporal choices, meaning it may already “know” what human-like behavior looks like. In contrast, the advantage of Arithmetic-GPT lies in our complete control over its training, eliminating data contamination and achieving comparable predictive accuracy to off-the-shelf LLMs. This control allows cognitive scientists to systematically investigate which computational tasks (used as training data) lead to human-like representations in language models. In this case, we have shown that just training on arithmetic is sufficient to produce good performance.
>
> **Question 1: Directly training LLaMA or Arithmetic-GPT on task features as an alternative.**
>
> **Response:** Thank you for this question. The choice depends on the specific scientific question. Task features serve as an abstract representation of stimuli; in risky choice experiments, for example, only probabilities and values—deemed essential by experimenters—are included. If the goal is to predict human behavior based on text descriptions of psychological tasks, fine-tuning off-the-shelf LLMs on human data (formatted as text descriptions of the task that include task features) may be effective; Centaur, for instance, is a model that follows this approach [1]. However, if the aim is to estimate the maximal explainable variance, training an MLP on task features, which represent experimentally manipulated variables among stimuli, is often sufficient. In our study, we use Arithmetic-GPT to explore the computational problems that people need to solve. For this purpose, directly using task features to predict human data is less informative, as it overlooks the computational problem we aim to understand.
>
> [1] https://arxiv.org/abs/2410.20268

---

> > ### Comment · Reviewer_eVJL · 2024-11-28
> >
> > Thank you to the authors for addressing my questions. I will update my score accordingly.

---

### Official Review · Reviewer_EZUX · 2024-11-04

**Soundness:** 3
**Presentation:** 3
**Contribution:** 3
**Rating:** 6
**Confidence:** 4

**Summary:**

The paper presented a method to enable a model to act as a cognitive model for a cognitive task by training it on 1) computationally similar tasks 2) with similar distribution. The authors demonstrate their method on the task of risky and intertemporal choice, by pre-training on synthetic expected value calculations data of ecological distribution. The paper compares their trained model, Arithmetic-GPT, with Llama-3-70B, and a few traditional cognitive models. Arithmetic-GPT shows a good ability to explain human data.

**Strengths:**

1. The paper is clear and well-written.
2. The method is novel and demonstrates strong results for a small model and relatively small data.
3. The results showcase the method's ability to create a good cognitive model for the target task.
4. The deduction of implicit values from ARITHMETIC-GPT is interesting and insightful.

**Weaknesses:**

1. The paper is a bit lean on the experimental side.
2. Lack of experimentation with different model sizes that can allow a better understanding of their effect on the model's ability to act as a cognitive model. Similarly, there are no experiments designed to assess the effect of the data quantity that is emphasized as key factors that differentiate humans from LLMs.
3. The paper claims that their approach is general, but they only train on one type of computational task, e.g. expected value calculations. Another example can enhance the paper, even pointing to other concrete examples can be valuable. For example, does this also hold to more language-oriented tasks? in that case, are smaller dedicated models will also outperform strong LLMs?
4. Part of your rationale for choosing ecological distribution is that in humans this can lead to computational errors that are the basis for deviating from the rational decision. However, you didn't analyze both LLaMA3 and Arithmetic-GPT computational abilities both in and out of distributions, and examine their effect on the results.
5. There is no discussion on the trade-off between a general model like LLaMA3 that can act as a cognitive model in a wide array of tasks and a small and dedicated model as you proposed.

**Questions:**

1. There are open models with open data e.g. LLM360 and OLMo, how does this affect your LLMs experiment in the context of the paper experiments?
2. Is there a way to produce figures like the ones in Figure 2 from LLaMA3 embeddings?
3. Can a strong model trained on the human data provide a better upper limit and also act as a cognitive model?

Comments:
1. LLaMA3 is a bit better in Intertemporal choices and on par with the Arithmetic-GPT. The text there doesn't reflect that and does not explain it. Also, the top results from LLaMA3 are not in bold.

---

> ### Author Response · Authors · 2024-11-22
>
> **Weakness 1 & 2: Lack of experiments that manipulate model sizes and data quantity for Arithmetic-GPT.**
>
> **Response:** We appreciate the reviewer’s suggestion and have conducted an additional analysis to systematically examine the effects of model size and data quantity. Specifically, we considered three model sizes (10M, 1M, and 30K parameters) by varying the hidden size of Arithmetic-GPT. Independently, we analyzed the impact of three dataset sizes (1M, 100K, and 10K mathematical equations), with the smaller datasets being random subsets of the original 1M ecological synthetic dataset. The detailed results are presented in Table A3 of the revised manuscript. In summary, we observed that larger model sizes consistently improve performance, while reducing the dataset size to as few as 10K equations has only a modest impact on performance.
>
> **Weakness 3: Lack of other computational tasks besides expected value calculation. Does the result hold for language-oriented tasks?**
>
> **Response:** Thank you for these suggestions. We believe our ablated synthetic datasets, where expected values were removed from the right-hand-side of the equations, offer some insight into alternative computational tasks. In these cases, the models trained on ablated data could not learn to perform expected value calculations properly but may still capture certain statistical patterns, such as frequencies and associations between probabilities and values. This led to decreased performance, supporting the role of explicit expected value calculations. However, the reviewer’s suggestion of structurally different tasks beyond simple ablations is interesting. While theoretically, there are countless computational tasks that could be used to train language models, identifying which tasks yield meaningful hypotheses is challenging. In our study, we focused on expected value calculations due to their direct relevance to rational decision-making. We believe that in other domains, such as language tasks, training on rational solutions could serve as a valuable first step.
>
> **Weakness 4: Evaluate LLaMA and Arithmetic-GPT’s computational abilities both in and out of distributions, and examine their effect on the results.**
>
> **Response:** Thank you for this comment. We would appreciate further clarification to provide a more precise response. If the reviewer is referring to the effect of training data distribution on model performance, we addressed this in the paper by comparing Arithmetic-GPTs trained on data following either a uniform distribution or an ecological distribution. Our results demonstrate that training with ecological distributions leads to improved performance compared to uniform distributions. We have also included an additional exploratory analysis in the newly added Appendix F, titled “Variations in Model Sizes, Data Quantity, and Data Distributions.”
>
> **Weakness 5: No discussion of the tradeoff between more general models (like LLaMA) vs. specific models (like our Arithmetic-GPT)**
>
> **Response:** This is a great point, thank you! In response, we have added the following to the Discussion section:
> “In contrast, we explicitly manipulate the training data for our Arithmetic-GPT, thereby uncovering a key factor that contributes to the model's ability to explain human choices. This targeted approach, however, comes at the cost of the broader versatility inherent in off-the-shelf LLMs, which are designed to handle a wider range of tasks.”

---

> ### Author Response · Authors · 2024-11-22
>
> **Question 1: How do open-source LLMs like LLM360/OLMo perform on our task?**
>
> **Response:** Thank you for this question. One of our priorities in selecting LLMs is to control for potential data contamination—specifically, whether models were trained on test questions. We examined open-source LLMs, such as OLMo (trained on the Dolma corpus [1]) and GPT-J (trained on the Pile [2]), both of which include scientific literature on human biases in risky and intertemporal choices, dating back several decades. Due to this overlap, these models do not provide additional control benefits over other open-weight models like LLaMA. As a result, we have opted to leave this comparison for future research.
>
> [1] https://arxiv.org/abs/2402.00159
>
> [2] https://arxiv.org/abs/2101.00027
>
> **Question 2: Visualizing LLaMA3’s embedding for values, probabilities, and discount factors.**
>
> **Response:** Thank you for this excellent suggestion. We visualized the embeddings of LLaMA3-70B-Instruct for these quantities and observed that only the discount factor exhibits the hyperbolic discounting curve commonly found in behavioral economics. We have included this result in the revised manuscript Fig A3.
>
> **Question 3: Can a strong LLM fine-tuned on human data offer an upper bound on model performance?**
>
> **Response:** We believe that within specific domains of human cognition, an MLP trained directly on human data sets the upper bound for capturing maximal explainable variance. For example, Binz & Schulz (2024) fine-tuned the LLaMA-1-65B model on the choices13k dataset and achieved a decision preference accuracy of 81.9% (see footnote 2 in pg. 15 of [3]), whereas the original choices13k paper reported that the best-performing MLP model achieved an accuracy of 84.15% [4]. Decision preference accuracy is defined as the proportion of decision problems where the model-predicted probabilities align with the empirical choice probabilities on the same side of 0.5. Therefore, fine-tuning off-the-shelf LLMs may achieve similar predictive accuracy; however, like MLPs, these fine-tuned models offer limited interpretability regarding what enables their strong predictive performance. Gaining control over the LLMs’ training data would allow cognitive scientists to better examine which data contribute to more human-like representations in LLMs.
>
> [3] https://arxiv.org/abs/2306.03917
>
> [4] https://doi.org/10.1126/science.abe2629
>
> **Comment 1: Add texts to reflect that LLaMA3 is on par with the Arithmetic-GPT in predicting intertemporal choice. And bold best-performing models within each of the categories.**
>
> **Response:** Thank you for the suggestion. We agree and will update the tables and text to bold the best-performing models within each category. Additionally, we have clarified in the text that LLaMA3 performs comparably to Arithmetic-GPT in predicting intertemporal choice.

---

> > ### Comment · Reviewer_EZUX · 2024-11-26
> >
> > Question 1:
> > I understand your perspective. In those cases, the problem seems less severe, as there are open-data contamination tests and tools available. For instance, tools like "What is in my big data?" can assess contamination levels and, more importantly, exclude contaminated examples from testing.
> >
> > More broadly, while you emphasize the lack of disclosed data and include a paragraph on "The Importance of Training Data Disclosure," the core issue appears to be contamination (discussed in the paper). Yet, contamination affects all models to the extent described here. Thus, I believe that the focus on nondisclosure is disproportionately emphasized in your paper relative to the role it actually has on the usability of those LLMs as cognitive models.
> >
> > Question 3:
> > Regarding the upper-bound discussion, I respectfully disagree. Specifically, Llama3 outperformed the MLP in the "Intertemporal choices" task, and assuming that supervised training could further improve its results, points towards the option that the MLP does not provide a relevant upper bound. Additionally, as observed in your experiments with smaller models and datasets, changes in data and model size can produce unexpected outcomes. In this case, I suspect that using 13K samples for a 65B model might be suboptimal, potentially allowing a smaller model to achieve better performance.

---

> > > ### Author Response · Authors · 2024-11-28
> > >
> > > Thank you for the additional comments.
> > >
> > > **Weakness 1 & 2:** Thank you for noting that our experimental work is sufficient. We believe the results from smaller datasets suggest that 10k examples from ecological synthetic datasets are indeed adequate to obtain an embedding capable of predicting human choices. It is also important to note that we did not train Arithmetic-GPT directly on the four human datasets. Therefore, we can confidently state that there is no overfitting to human data, as the model was trained solely to perform arithmetic. To clarify this point, we have added the following text:
> > >
> > > “The results, summarized in Table A3, reveal an intriguing pattern: reducing the size of the ecological synthetic dataset to as few as 10K equations does not significantly impair the ability of the 10M model’s embeddings to predict human choices. It is important to note that Arithmetic-GPT models were never exposed to human data during pretraining.”
> > >
> > > **Weakness 3:** It is indeed challenging to identify a meaningful computational task that produces a human-like embedding when a language model is trained on it. We addressed this challenge by identifying one such task – expected value calculation – and demonstrated that human-like embeddings emerge for risky and intertemporal choices, as explicitly stated in our title. Through this process, we discovered that performing expected value calculations on ecological distributions is crucial, allowing a language model to significantly narrow the performance gap with MLPs in these domains. This is an advantage that open-weight LLMs do not currently provide, whereas open-source LLMs might be able to offer through sophisticated interpretability techniques.
> > >
> > > Our goal is not to outperform off-the-shelf LLMs across all domains or MLPs in specific domains. Instead, our aim is to use language models as a tool to uncover useful computational tasks that can help explain human cognition in certain contexts. We believe this approach has broad applicability to many areas of cognitive science, particularly where clearly defined computational tasks for cognitive processes are still lacking.
> > >
> > > **Weakness 4:** Thank you for the clarification. In response, we generated two new datasets for expected value calculations – one with uniform distributions and another with ecological distributions – to evaluate the arithmetic abilities of Arithmetic-GPT and LLaMA-3. Detailed results are presented in the newly added Appendix G, titled “Computational Capabilities of Language Models.” Our findings suggest that a language model’s ability to compute expected values in the ecological test set may help predict the ranking of its embeddings’ predictive power for human choices. However, we also emphasize a caveat: directly relating a model’s arithmetic performance (i.e., model behavior) to the predictive power of its embeddings requires caution and highlights the need for further research in this area.
> > >
> > > **Weakness 5:** We respectfully disagree with the assumption regarding LLaMA’s training process. First, to the best of our knowledge, the details of LLaMA’s training process are not publicly available. While it is unlikely that LLaMA underwent the same specialized pretraining as Arithmetic-GPT, it is plausible that, during its training on Internet-scale text, LLaMA was exposed to similarly ecological datasets involving expected value calculations. Moreover, there is a possibility that LLaMA’s training corpus directly included the four human datasets (i.e., choices13k, cpc18, gershman20, and agrawal23) or psychological research papers discussing human biases in risky and intertemporal choices. If this were the case, it would not be surprising for LLaMA to outperform our model, as it may have been exposed to the test questions or their underlying human biases during training. Therefore, we agree that large and powerful models may excel at predicting human choices in certain domains (especially when they already have been trained on these domains). However, without thoroughly addressing the issue of potential data contamination, we believe these larger models cannot yet be considered superior cognitive models that advance our understanding of human cognition.

---

> > > > ### Author Response · Authors · 2024-11-28
> > > >
> > > > **Question 1:** We agree that open-source LLMs are more valuable for scientific purposes than open-weight LLMs, primarily because, as the reviewer mentioned, effective scientific inference requires transparency regarding an LLM’s training data and process. In response to this point, we have made the data contamination issue more explicit in the section “The Importance of Training Data Disclosure”:
> > > >
> > > > “This is primarily because their training data is rarely disclosed, making it difficult for scientists to control for data contamination and thereby precisely identify the sources of human-like behaviors in these models.”
> > > >
> > > > **Question 3:** It is true that, in certain cases, LLaMA3 can marginally outperform MLP, even when MLP is directly trained on human data. In our original submission, we discussed this between lines 406 and 410: “Except for the intertemporal choice tasks, MLPs outperform all other models. It is important to note that MLP training was based on task features rather than text descriptions that simulated participants’ experiences or arithmetic equations that mimic a rational agent’s computations. Consequently, these differences in input formats could also lead to diverging performance.”
> > > >
> > > > This suggests that there may be additional variance in human data that cannot be fully captured by task features but is instead present in text descriptions. To push this idea further, one could imagine a more powerful model receiving an exact replication of the task as input—this would include task features, text descriptions, stimuli displays on participants’ computer screens, and even environmental factors like lighting conditions. In such a scenario, the model could perfectly replicate the participant’s experience when performing the cognitive tasks. However, cognitive models must inherently operate at certain levels of abstraction, which inevitably introduces limitations and imperfect comparisons, particularly in the era of LLMs.
> > > >
> > > > It could also be argued that the 13k examples from the choices13k dataset are insufficient, as contextual factors (elements influencing choices beyond task features) may remain significant, necessitating larger datasets. However, for the choices13k dataset, it is worth noting that MLP consistently outperforms both LLaMA3 and Arithmetic-GPT, serving as a reasonable upper bound for this dataset.
> > > >
> > > > We believe the reviewer intended to highlight that the 10k examples from the largest intertemporal choice dataset (agrawal23) might be insufficient, as LLaMA3-70B-Instruct outperforms MLP on this dataset. In response, we have added the following text to help readers better interpret the results from intertemporal choices:
> > > >
> > > > “We observe that LLaMA3 can outperform MLP on agrawal23. While the two neural network models use different task formats as inputs, which may contribute to the differences in performance, it is also possible that the dataset lacks sufficient statistical power to reliably distinguish between the two models.”

---

> > > > > ### Comment · Reviewer_EZUX · 2024-11-28
> > > > >
> > > > > Question 3: The main point was that the MLP upper bound is a reasonable upper bound for the "Classical behavioral models" but when you consider a stronger model as a cognitive model it makes sense to adjust the class of upper bound models you consider accordingly. For example, to train an A-GPT-like model on the human data as upper-bound.
> > > > > It is clear that considering a wider class of stronger models is expected to yield a higher upper bound.
> > > > > (the last comment tried to demonstrate that)
> > > > > Any result that is higher than an upper bound is odd (even marginally), and calls to question the setup that facilitates it.
> > > > > This statement combined with the first assertion suggests that the current upper bound for "Intertemporal choices" needs to be adjusted.

---

> > > > ### Comment · Reviewer_EZUX · 2024-11-28
> > > >
> > > > Thank you for your replay:
> > > > Weakness 1 & 2: I did not say that your training was overfitted over the human data but rather overfitted on the train data, and for that 10k was sufficient. Hence more data from the same distribution did not further improve the results.
> > > >
> > > > Weakness 3: I did not undermine your objective to provide a small controlled llm-based cognitive model, but merely express concern regarding the outlined approach for language-oriented tasks. Still, I think we can agree that it is completely reasonable to leave this aspect of the work for further research.
> > > >
> > > > Weakness 4: I recognize the effort involved in running such an experiment in such a short time. Reding App. G, I noticed you evaluated the A-GPT differently from the way you evaluated the Lama-3 model. This discrepancy is what I suspect caused the big difference in performances and makes the comparison across those experimental setups not valid. Nonetheless, it is interesting to see the effect of the training distribution of the model behavior and its connection to its ability to facilitate prediction over human behavior.

---

> ### Comment · Reviewer_EZUX · 2024-11-26
>
> Thank you for your response.
>
> Weaknesses 1 & 2:
> I found the experiments sufficient overall. While you explained the importance of model capacity, you did not address why the smaller dataset size did not affect performance. The lack of a performance drop suggests that your model might overfit to the target data distribution. This connects to Weakness 3 and the comment about LLaMA3, as it highlights how overfitting to the task and domain is not always straightforward, yet appears to be the case here.
>
> That said, while not strictly necessary for this study, it would be interesting to explore the effects of using larger models or incorporating modifications like increased model depth. Such experiments could even provide an initial direction toward scaling laws for cognitive models.
>
> Weakness 3:
> As you mentioned, connecting computational tasks to meaningful hypotheses is challenging. This poses a limitation when presenting your approach as a general one. For language-oriented tasks, I tend to think that training on rational solutions, while valid as a first step, may not yield a cognitive model that is on par with larger, stronger models—particularly within the model size class discussed in the paper (putting aside the problems of using LLM as a cognitive model).
>
> Weakness 4:
> Apologies for the earlier unclear comment. I intended to suggest evaluating LLaMA's and Arithmetic-GPT’s computational abilities more explicitly—specifically, benchmarking their expected value calculation abilities both in- and out-of-distribution. This could provide insights into their connection to the results. For instance, does a model that excels in expected value calculations also serve as a better cognitive model? Are 15K examples enough to master this ability?
>
> Weakness 5:
> I’m not sure my earlier point came across clearly. My point was that LLaMA models perform comparably to your model for the given task, despite not being trained specifically for it. When combined with your statement about the difficulty of finding computational tasks that improve cognitive modeling for target tasks, this suggests that for many use cases, using a larger, stronger general-purpose model may be more effective than training a specialized cognitive model.
>
> Overall:
> I find your paper interesting and thought-provoking. However, I feel that the presentation of your approach as a general approach (e.g., in the abstract) is somewhat overstated compared to its actual applicability.

---

### Official Review · Reviewer_2CbN · 2024-11-04

**Soundness:** 3
**Presentation:** 2
**Contribution:** 3
**Rating:** 8
**Confidence:** 3

**Summary:**

The paper shows that by solely training next token prediction on arithmetic data requiring computations related to real-world decisions (e.g., computing expectations) it's possible to predict with good accuracy human behavior. Results improve (slightly?) when the pretrained computations involve numbers which resemble real-world distributions.

**Strengths:**

- I found the results of the paper very surprising, I think that the approach is creative, and to the best of my knowledge, may perhaps offer a minimal explanation for why LLMs replicate human behavior in some cognitive biases - namely, something about the next word prediction mechanism and mathematical data? Without the need for any language-related observations at all.

**Weaknesses:**

I don't have strong concerns, and I'm generally positive about this paper, though I feel like Section 3.4 (Human Targets) contains many preprocessing decisions with some very specific factors (annual discount factor set to 0.85). I didn’t understand the need for these, or whether they affect the results. For example, I don’t understand this part: “In cases involving ambiguous gambles where probabilities are unknown, we used the special token to denote gambles with unknown probabilities.” (Line 237) Can you please elaborate?

**Questions:**

- I wonder if the positioning of the paper may be different - something along the lines of "LLMs were observed to replicate human cognitive biases, here we show a possible explanation, indicating that this may be due to the task, architecture, and numerical reasoning, without requiring any natural language supervision, world knowledge or common sense." Does this make sense? If so, I think it would have been easier for me to understand the paper.
- I wonder what would happen if the LLM was trained on other mathematical tasks, not relating to expectation? Would it still be able to explain human behavior to some extent?

---

> ### Author Response · Authors · 2024-11-22
>
> We thank the reviewer for the positive evaluation of our paper.
> **No weakness was identified in our work. Clarify the need to specify the discount factor and token for ambiguous probabilities.**
>
> **Response:** The choice of an annual discount rate of 0.85 follows typical equity discount rate in the real world. We imposed this constraint on our Arithmetic-GPT model to prevent other cognitive mechanisms (e.g., noisy discounting factors) from artificially enhancing the model’s ability to produce more human-like deviations from rationality. By setting a fixed discount rate, we enhance the robustness of our results, ensuring that observed effects are intrinsic to the model rather than artifacts of other non-PV calculations.
>
> Similarly, in cases involving risky choices where probabilities are unknown, such as when participants are offered an option of 100 dollars with unspecified probabilities, we address ambiguity by defining the expected value of this option as <AMB> * 100. Here, <AMB> is a token explicitly designed to capture situations of ambiguous probability, allowing for a systematic treatment of ambiguous probabilities replicating how human participants might perceive such options.
>
> **Question 1: The positioning of the paper can be improved.**
>
> **Response:** Thank you for this excellent suggestion. In response, we have revised parts of the Introduction and Discussion sections as follows:
> Introduction:
> “We propose a hypothesis about how human-like decision patterns might be produced in language models for risky and intertemporal choice: such human-like behaviors might arise from a language model trained on ecologically valid calculations of expectations. This suggests that human-like biases could result from the training task and numerical reasoning, independent of natural language supervision.”
>
> Discussion:
> “This pretraining allows the model to better capture human risky and intertemporal
> choices compared to classical behavioral models and, in some cases, even surpass the performance of the general-purpose LLaMA3 model. These results suggest that the observed cognitive biases in LLMs may stem from the training task, architecture, and numerical reasoning abilities, without requiring natural language supervision.”
>
> **Question 2: LLM’s training on other mathematical tasks not related to expectation.**
>
> **Response:** Within the task of expectation calculation, we observe that the distributions of probabilities and values in the training data significantly influence the representations learned by the language model. Moreover, ablating the expectation calculation task itself leads to a decrease in model performance, indicating the centrality of this task to the model’s success. These two findings suggest that modifying tasks—whether by changing task content (e.g., from expectation calculation to other types of task) or by adjusting the training distributions (e.g., from uniform to ecological distributions)—would likely yield different model behaviors.
>
> In evaluating our models on four distinct human datasets, we find that the embeddings generated by Arithmetic-GPT are nearly comparable to those of an MLP directly trained on human data, with only a 1-4% difference in $R^2$. This suggests that training on the expectation calculation task has likely brought Arithmetic-GPT close to ceiling performance, further underscoring that deviating from training on expectation tasks likely leads to less correspondence with human behavior.

---

### Author Response · Authors · 2024-11-22

We sincerely thank all the reviewers for their thorough evaluation of our paper and their constructive feedback. We believe that these revisions have significantly enhanced the quality of the paper. Below, we address each comment in detail. Here, we summarize the key changes and additional analyses incorporated into the revised manuscript:
- **New Experimental Results on Model Sizes and Data Quantity:** We conducted experiments varying the model sizes of Arithmetic-GPT and the amount of pretraining data (see Table A3 in Appendix F, “Variations in Model Sizes, Data Quantity, and Data Distributions”).
- **New Experimental Results on Data Distributions:** We explored the effects of varying data distributions on model performance (see Table A4 in Appendix F, “Variations in Model Sizes, Data Quantity, and Data Distributions”).
- **Additional Analysis of LLaMA-3-70B-Instruct Embeddings:** We included an analysis of embeddings derived from LLaMA-3-70B-Instruct (see Figure A3 in Appendix E, “Additional Implicit Functions”).

In addition to these updates, we refined the text throughout the manuscript to align with the reviewers’ suggestions and the new findings. In the revised manuscript, changes are highlighted in blue for ease of review.

---

### Meta-Review · Area_Chair_H1DQ · 2024-12-23

**Metareview:**

The paper examines how Arithmetic-GPT, a small model pretrained on ecologically valid arithmetic datasets, aligns with human decision-making in risky and intertemporal tasks. It outperforms traditional models and rivals larger LLMs like LLaMA-3-70B in specific cases. The work’s strengths include its novelty, rigorous benchmarking, and interpretable methodology. However, the scope is narrow, focusing only on expected value calculations, and the gains over LLaMA-3-70B are modest. Claims of generalizability lack support from broader experiments.

**Additional Comments On Reviewer Discussion:**

Reviewers praised the innovative approach and clarity but raised concerns about the narrow focus and modest gains. Authors addressed some issues with additional experiments and clarifications on ecological datasets and training trade-offs. However, doubts about generalizability and broader cognitive applications remain.

---

### Decision · Program_Chairs · 2025-01-22

Accept (Poster)